# Chitin Nanocrystal Hydrophobicity Adjustment by Fatty Acid Esterification for Improved Polylactic Acid Nanocomposites

**DOI:** 10.3390/polym14132619

**Published:** 2022-06-28

**Authors:** Ivanna Colijn, Murat Yanat, Geertje Terhaerdt, Karin Molenveld, Carmen G. Boeriu, Karin Schroën

**Affiliations:** 1Food Process Engineering Group, Wageningen University and Research, Bornse Weilanden 9, 6708 WG Wageningen, The Netherlands; murat.yanat@wur.nl (M.Y.); geertjeterhaerdt@live.nl (G.T.); karin.schroen@wur.nl (K.S.); 2Wageningen Food and Biobased Research, Wageningen University and Research, Bornse Weilanden 9, 6700 AA Wageningen, The Netherlands; karin.molenveld@wur.nl (K.M.); carmen.boeriu@wur.nl (C.G.B.)

**Keywords:** chitin nanocrystals, Steglich esterification, fatty acids, nanocomposites, polylactic acid, surface acetylation

## Abstract

Bioplastics may solve environmental issues related to the current linear plastic economy, but they need improvement to be viable alternatives. To achieve this, we aimed to add chitin nanocrystals (ChNC) to polylactic acid (PLA), which is known to alter material properties while maintaining a fully bio-based character. However, ChNC are not particularly compatible with PLA, and surface modification with fatty acids was used to improve this. We used fatty acids that are different in carbon chain length (C4–C18) and degree of saturation (C18:2). We successfully used Steglich esterification and confirmed covalent attachment of fatty acids to the ChNC with FTIR and solid-state ^13^C NMR. The morphology of the ChNC remained intact after surface modification, as observed by TEM. ChNC modified with C4 and C8 showed higher degrees of substitution compared to fatty acids with a longer aliphatic tail, while particles modified with the longest fatty acid showed the highest hydrophobicity. The addition of ChNC to the PLA matrix resulted in brown color formation that was reduced when using modified particles, leading to higher transparency, most probably as a result of better dispersibility of modified ChNC, as observed by SEM. In general, addition of ChNC provided high UV-protection to the base polymer material, which is an additional feature that can be created through the addition of ChNC, which is not at the expense of the barrier properties, or the mechanical strength.

## 1. Introduction

To tackle environmental issues related to the use and production of fossil-based plastics, bio-based and biodegradable plastics have been proposed as alternatives. Amongst them, polylactic acid (PLA) is often considered the most promising material because of its availability, low environmental footprint, low costs, good optical properties, and high tensile modulus [1]. However, its current application is limited because its properties are not as good as its fossil-fuel-based counterparts, such as polyethylene terephthalate [2,3].

Nanoparticle–polymer interactions are known to alter material properties including mechanical and barrier functions [4,5,6,7,8]. To retain a plastic’s biobased character, the particles also need to comply with this requirement, and different polysaccharide nanoparticles have been considered for this purpose such as starch or cellulose nanoparticles, that are abundant in nature, and have low toxicity [9]. Another promising nanoparticle source is chitin, which is a polysaccharide composed of N-acetyl-2-amido-2-deoxy-D-glucoside units linked by β(1→4) bonds. After cellulose, it is the second most abundant polysaccharide and mainly present in exoskeletons of arthropods such as shrimps and crabs.

Currently, these exoskeletons are considered waste, which is a missed opportunity as chitin can be easily extracted and made into a high-added-value product. Chitin nanocrystals (ChNC) can be produced by acid hydrolysis of chitin powder. Like the chitin powder, the degree of acetylation (presence of amino groups) at the nanocrystal surface is thought to relate to antifungal and antibacterial activity, even when ChNC are present in polymer films [10,11,12], which is of special interest for medical devices or food packages [13,14]. ChNC clearly have high potential as building blocks in nanocomposites, albeit they are rather unexplored.

For a functional biopolymer, homogeneous distribution of ChNC throughout the PLA film is crucial [15]. When starting from ChNC, the relatively hydrophobic PLA drives the ChNC to agglomerate. During particle preparation, interparticle hydrogen bonds lead to strong ChNC agglomerates upon drying [16] which are difficult to break up under conditions commonly used during extrusion processes. In order to mitigate this, ChNC can be modified, which is possible through their hydroxyl and amino groups [17,18].

Surface acetylation with fatty acids has been shown to effectively increase the hydrophobicity of starch crystals [19] and chitin nanocrystals [17]. Commonly, this is achieved by converting the fatty acids into their noxious chloride forms, which is less desirable from a human health and environmental point of view. Alternatively, direct esterification (Steglich esterification) that circumvents formation of toxic components and is efficient under mild conditions can be used, but to the best of our knowledge, this has not been demonstrated for chitin nanocrystals. Here, we use fatty acids of different carbon chain length (C4–C18) and saturation (C18:0 and C18:2), and characterize the modified particles by FTIR and solid-state NMR. We expect the polarity to depend on the length of the carbon tail, and this is investigated using two-phase-liquid-systems of various *logP*. Next, (modified) ChNC were added to a polymer matrix, and dispersibility was investigated through SEM. Furthermore, the characteristics of the obtained nanocomposites were evaluated (e.g., UV protection, color, mechanical strength, barrier function). This modification is expected to open up possibilities to successfully use chitin nanocrystals as fillers for bio-nanocomposites.

## 2. Materials and Methods

### 2.1. Materials

Shrimp chitin powder (≥98% purity) was purchased from Glentham Life Sciences (Corsham, UK). Butyric acid (≥99.5%) was from Thermo Fisher Scientific (Waltham, MA, USA). The following chemicals were bought from Sigma Aldrich (St. Louis, MO, USA): octanoic acid (≥99% purity), lauric acid (≥98% purity), stearic acid (99% purity) linoleic acid (≥99% purity), N-(3-dimethylaminopropyl)-N′-ethyl carbodiimide hydrochloride (EDC), hexane tert-butyl acetate (≥99% purity), and 4-dimethylaminopyridine (DMAP). Palmitic acid (≥99% purity) and dry tetrahydrofuran (THF) (≥99% purity) were purchased from Merck (Darmstadt, Germany). Polylactic acid Ingeo 4043D was from NatureWorks LLC (Plymouth, MN, USA). All other chemicals and solvents were analytical-grade and used as received. For dilutions, only ultrapure water was used (Q-POD with Millipak Express 40 0.22 µm filter, Merck Millipore, Burlington, MA, USA).

### 2.2. Sample Preparation

#### 2.2.1. Chitin Nanocrystals Preparation

A general acid hydrolysis procedure was used to prepare chitin nanocrystals, as described in [16,20]; the effects of production conditions on nanocrystal properties are extensively reviewed in [18]. Crude chitin powder was hydrolyzed in 3 M HCl for 90 min at 90 °C, after which the reaction was stopped by cooling the mixture on ice. To remove the HCl, the mixture was centrifuged at 4000× *g* for 5 min, after which the supernatant was discarded, and the pellet redispersed in water. The latter steps were repeated three times. This suspension was sonicated with the Branson Sonifier SFX550 (Brookfield, CT, USA) equipped with a sonication tip 1/8′ tapered microtip (Branson, Brookfield, CT, USA), in pulses of 100 J at an amplitude of 40% with 10 s rest, and a total sonication energy of ~150 J/mL while cooled on ice. To collect the chitin nanocrystals (ChNC), two centrifugation steps were applied (1000× *g*, 15 min), after which the pellet was discarded.

#### 2.2.2. Surface Acetylation

Before modification, the ChNC and the chemicals were dried in a desiccator with silica for 5 days. The ChNC had a water activity (aw) ~0.25. A dispersion of 5.0 *w*/*v*% freeze-dried ChNC was made in THF and stirred >24 h at room temperature. This suspension was sonicated in pulses of 100 J at an amplitude of 40% with 10 s of rest and a total sonication energy of ~ 150 J/mL while cooled on ice. After sonication, the sample was kept on ice and the fatty acids (1 fatty acid: 5 chitin monomers) and DMAP (3 DMAP: 1 fatty acid) were added. EDC was slowly added, and the mixture was stirred for 5 more minutes after addition. The reaction was started by increasing the temperature to 45 °C. The mixture was kept at this temperature for 45 min, and the reaction was stopped by centrifugation at 4700× *g*. The pellet was rinsed one more time with THF, followed by two rinsing steps with methanol and acetone. The pellet was dried in a vacuum oven (VD53, Binder, Tuttlingen, Germany) under constant nitrogen flow at 40 °C for >24 h. The supernatants containing the unbound fatty acids, EDC, and DMAP were pooled and saved for further analysis with GC-FID. All modifications were performed in independent duplicates. Washed ChNC refer to ChNC that were subjected to the same washing steps as the modified ChNC, whereas unwashed ChNC refer to ChNC that were used directly after production.

### 2.3. Nanocomposite Production

Prior to production, PLA 4043 was dried in a desiccant air dryer (TTM 2/100 ES, Gerco Kunststofftechnik GmbH, Ennigerloh, Germany) at 80 °C for >48 h and the (modified) ChNC were dried at 50 °C in a vacuum oven; this was performed under continuous nitrogen flow for >48 h while allowing minimal vapor to escape the oven. Thereafter, a micro-compounder (MC 15 HT, Xplore, Sittard, The Netherlands) was used to produce nanocomposites consisting of 95 wt.% PLA and 5 wt.% (modified) chitin nanocrystals at 180 °C. During sample addition, a maximum torque of 40 Nm and screw speed of 40 rpm/s^2^ were used. During compounding, the screw speed was increased to 100 rpm/s^2^, and mixing was continued for approximately 2 min. After extrusion, samples were pressed into sheets using a hot press (LabEcon 600, Fontijne Presses, Delft, The Netherlands) at 190 °C; a pressure of 10 kN was applied for the first three minutes followed by a pressure of 50 kN for two minutes. An in-house-made mould from aluminum was used to control the sample thickness at ~400 µm. The pressed samples were cooled for 2 min in an in-house-made cooling unit consisting of two plates that were continuously cooled by rinsing tap water.

### 2.4. Nanoparticle Characterization

#### 2.4.1. TEM

Transmission electron microscopy (TEM) images were taken using JOEL-JEM1400Plus–120 kV (spot size 1). Before observation, the ChNC were negatively stained with a 2% uranyl acetate solution. Three images per sample were further analyzed with Fiji [21]. To determine particle diameter, length, and aspect ratio, 52 ChNC were analyzed per image.

#### 2.4.2. FTIR

FT-IR spectra of (un)modified ChNC were obtained using a Bruker Equinox 55 (Munich, Germany) in attenuated reflectance mode (400–4000 cm^−1^), with a resolution of 4 cm^−1^ and after 64 accumulations. Calibration and baseline placement of [22] was applied to determine the degree of acetylation, using the ratio between amide II (1560 cm^−1^) and glycosidic bond (1030 cm^−1^) for quantification.

#### 2.4.3. Solid-State ^13^C NMR

The ^13^C cross-polarization magic angle spinning CP-MAS NMR spectrum was obtained on a Bruker Avance III HD spectrometer operating at 700.13 MHz (16.4 T). All particles were packed into 4 mm zirconia rotors that were spun at MAS frequency of 11 kHz at 298 K. The ^13^C CP MAS spectra were recorded with a recycle delay of 5 s, and contact time of 3 ms. The ^13^C NMR spectra were referenced with respect to adamantane (^13^C, 29.456 rpm). The spectra were analyzed using MestRenova software.

#### 2.4.4. GC-FID

The supernatant containing unbound fatty acid (with the exception of butyric acid) and the remaining EDC and DMAP was evaporated under constant nitrogen flow (Reacti-Therm III, Thermo Fisher Scientific, Waltham, MA, USA). Prior to analysis, fatty acid methyl esters (FAMES) were derived following a similar protocol as described in [23]. In short, the remaining components were redispersed in methanol (0.3 mL/mg fatty acid added for modification) and stirred overnight. Next, the samples were sonicated at an amplitude of 40% with a total energy input of 100 J/mL (SFX150, Branson, Brookfield, CT, USA). As reference, a known amount of pentanoic acid (C15:0) was added to each sample. An amount of 200 µL HCl in methanol (8:92 *v*/*v*%) was added per mL sample. FAMES were derived by heating this mixture at 90 °C for 1 h. The samples were cooled to room temperature, and subsequently 1 mL hexane and 1 mL MilliQ were added and vortexed. This was left to sit for 10 min, after which the hexane layer was taken and diluted 10 times prior to GC-FID analysis. For the quantification of unreacted butyric acid, a calibration curve of butyric acid in a mixture of acetone: tetrahydrofuran: methanol (1:1:1) was made. A known and equal amount of acetic acid was added as internal standard; this was added to the supernatant containing unreacted butyric acid and to each calibration curve point.

The unbound butyric acid and FAMES were quantified by gas chromatography (Focus GC, Thermo Scientific, Waltham, MA, USA) in combination with a flame ionization detector (Interscience, The Netherlands); the CP-FAAPCB column (Agilent, Santa Clara, CA, USA) and rxi-5 ms capillary column (Restek Corp, Bellefonte, PA, USA) were used for butyric acid and the FAMES, respectively. During butyric acid analysis, the oven was held at a temperature of 100 °C for 30 s, after which the temperature was increased to 180 °C with a ramp of 8 °C per minute. The temperature was kept at 180 °C for 1 min, after which it was increased to 200 °C with a ramp of 20 °C per minute. The sample was injected to the column (CP-FAAPCB, Agilent, Santa Clara, CA, USA) with a split flow of 40 mL per minute, while the oven was kept at 200 °C for 5 min. Nitrogen was used as carrier gas and applied at a constant pressure of 20 kPa. The detector was kept at a temperature of 240 °C. During FAMES quantification, the oven was held at a temperature of 40 °C for 2 min, after which the temperature was increased to 250 °C with a ramp of 10 °C per minute, and held at this temperature for 5 min. The temperature of the injector and detector was 240 °C and 250 °C, respectively.

The degree of substitution, DS, was calculated as follows:(1)$$\mathrm{Ds}=\frac{F_{\mathrm{added}}-{\;F}_{\mathrm{unbound}}}{4*\mathrm{ChNC}}*100$$
where *ChNC*, *F_added_*, and *F_unbound_* represent the amount of chitin added in moles, the amount of fatty acid added in moles, and the unbound fraction in moles, respectively. To calculate the total of available target groups, ChNC is multiplied by 4 to compensate for the hydroxyl groups of chitin.

#### 2.4.5. ζ-Potential

The ζ-potential of (un)modified ChNC in MilliQ was measured using Zetasizer Ultra (Malvern Panalytical, Malvern Ltd., Malvern, UK). Prior to analysis, the pH of these dispersions was adjusted to 5, and thereafter these were loaded into capillary cells (DTS1080, Malvern Panalytical, Malvern Ltd., Malvern, UK). All samples were measured in triplicate, where an absorption index was set to 0.01, and a refractive index of 1.56 and 1.33 was used for chitin nanocrystals and MilliQ water, respectively.

### 2.5. Nanocomposite Characterization

#### 2.5.1. Scanning Electron Microscopy

Neat PLA and nanocomposite films were cryo-fractured, glued on an aluminum sample holder with conductive carbon tape (Leit-C, Neubauer Chemikalien, Berlin, Germany), and sputter-coated with ~10 nm Tungsten (Leica EM SCD500, Leica Microsystems, Amsterdam, The Netherlands). The surfaces were observed with FESEM (FEI Magellan 400, FEI Electron Optics B.V., Eindhoven, The Netherlands) at room temperature at working distance of 4 mm, with SE detection at 2 kV and 13 pA.

#### 2.5.2. Color analysis

The color values (L*, a*, b*) of nanocomposite films were measured using a Minolta CR-400 colorimeter (Minolta Camera Co., Osaka, Japan). The colorimeter was calibrated with a standard white plate (D65, Y = 94.4, x = 0.3158, y = 0.3334) before use. L*, a*, and b* values were measured under D65 illumination. The measurement was performed in triplicate for all samples.

#### 2.5.3. Spectroscopy

The light transmittance of nanocomposite films was measured with a UV–VIS spectrophotometer (DU720, Beckman Coulter, Brea, CA, USA) in the range of 200–700 nm at room temperature. Instead of a cuvette, the pressed film was placed inside the sample holder of the UV–VIS, and air was used as background. Values were corrected based on film thickness.

#### 2.5.4. Barrier Properties

The water vapor transmission rate (WVTR) of PLA and nanocomposite films were determined according to ASTM E96. Samples were cut into circular films with a diameter of 3.8 cm, and fixed between an aluminum cup containing dry silica beads. During the measurement, the cups were placed in a conditioning chamber (PR-4J, Espec, Osaka, Japan) at 23 °C and 85% RH; the samples were approximately daily weighted with a four-digit analytical balance (ME204E, Mettler Toledo, Columbus, OH, USA) for a period of 14 days. The *WVTR* was calculated as follows:(2)$$WVTR=\frac{m}{tA}$$
where *m* is the water update by the silica beads (g), *t* is the testing time (days), and *A* is the surface area (m_2_) of the sample of choice.

#### 2.5.5. Mechanical Properties

Prior to analysis, a mould (DIEFAC stansvormen, Oosterhout, The Netherlands) was used to produce tensile test samples from the pressed sheets. After that, the samples were conditioned at a relative humidity of 50%, at 20 °C for 1 week. Tensile strength measurements were performed according to ISO 527-2 using a Zwick Z010 (Zwick Roell, Ulm, Germany); a clamp distance of 80 mm, an extensometer distance of 30 mm, an E-modulus speed of 1 mm/min, and a testing speed of 10 mm/min were used. The dimensions of the tensile strength measurement samples can be found in Appendix A Figure A1.

## 3. Results and Discussion

All tested fatty acids, butyric acid (C4:0), octanoic acid (C8:0), lauric acid (C12:0), stearic acid (C18:0), and linoleic acid (C18:2), were successfully coupled to chitin nanocrystals (ChNC) using Steglich esterification in the presence of E-ethyl-N’-carbodiimide (EDC) and 4-dimethylaminopyrine (DMAP). We first characterize the modified chitin nanocrystals, and thereafter the properties of PLA containing 5 wt.% (modified) chitin nanocrystals.

### 3.1. Nanoparticle Characterization

#### 3.1.1. Morphology and Size of (Modified) ChNC

Transmission electron microscopy was used to image all ChNC (Figure 1A). The length and diameter distribution as determined from three different regions in the sample are given in Figure 1B,C. Irrespective of the modification method used, ChNC particles are rod-like, with an average length of ~200 nm and diameter of ~10 nm, and thus have an aspect ratio of ~20. These dimensions are well within the ranges commonly found in literature, confirming successful chitin nanocrystal production [9,18,20]. In contrast to other esterification reactions where chloride intermediates were used [19,24], Steglich esterification did not affect the morphology or the size of the ChNC, thus confirming that Steglich esterification is rather mild.

#### 3.1.2. Degree of Substitution, Degree of Acetylation, and ζ-Potential

The FTIR spectra of (un)modified chitin nanocrystals can be found in Figure 2; the full spectra of all samples can be found in Appendix A Figure A2. The stretching behavior of unmodified ChNC corresponds well with values reported in literature [25]. The first broad peaks at 3430 cm^−1^ and 3258 cm^−1^ were assigned to the -OH and -NH stretch vibrations, respectively. The peaks at 1658 cm^−1^, 1628 cm^−1^, and 1563 cm^−1^ correspond to amide I, amide II, and amide III bands, which is typical stretching behavior of α-chitin [25]. The absorption bands between 1000 and 1200 cm^−1^ correspond to -C-O-stretching present in the polysaccharide backbone. An increased intensity was found upon modification in the region 2860–2900 cm^−1^ corresponding to the aliphatic chains of the fatty acids (CH_2_). In the region 1735–1750 cm^−1^, the formed ester linkages (C=O) appeared, with lower peak intensity for carbon chain lengths > 12.

Figure 3 shows the ^13^C NMR spectra of (un)modified ChNC; the full spectra of all samples can be found in Appendix A Figure A3. The spectra for unmodified ChNC show peaks corresponding to carbons C^1^–C^6^ (104.1 ppm, 55.3 ppm, 73.5 ppm, 83.5 ppm, 75.9 ppm, 61.1 ppm), the CH_3_ (C^8^, 22.9 ppm) and the acetyl group carbon (C^7^=O, 174.9 ppm). Modifications introduced new peaks in the region of 13.4–38.0 ppm; these could be assigned to the CH_3_ (25.3 ppm) and CH_2_ groups of the aliphatic tail of the fatty acids. Modification shifted the peak at 174.9 ppm to ~173 ppm, indicative of an ester link between the fatty acid and the ChNC (Table 1). Furthermore, the amide group present in the unmodified ChNC typically show a chemical shift at ~175 ppm, but the consistent chemical shift of 1.7 ppm at this wavelength indicates successful esterification.

Table 2 presents the degree of substitution determined with GC-FID, the degree of acetylation, and the ζ-potential of the (un)modified ChNC. Generally, the degree of substitution was ~2–4% upon modification, which did not affect the ζ-potential and had a minor effect on the degree of acetylation. In line with FTIR (Figure 2), GC-FID showed that the degree of substitution was higher for aliphatic tail lengths shorter than 12 carbons. This all indicates that the hydroxyl groups are modified and not the amine groups.

#### 3.1.3. Wettability Test

ChNC particles were added to two-phase systems consisting of MilliQ and either butanol (*logP* 0.88), tertbutyl acetate (*logP* 1.76), or toluene (*logP* 2.73) (Figure 4). Modification clearly changed the phase behavior of particles; the unmodified ChNC always migrated to the MilliQ phase irrespective of the organic solvent used, suggesting a rather polar character. The modified particles partitioned increasingly toward the organic phase as the length of the fatty acid increased, and ultimately rather fully accumulated in the solvent with highest *logP*. We expected all modified particles to have increased hydrophobicity compared to the starting material, which was also found. The fact that the degree of modification was lower for the longer chain fatty acids was apparently compensated for by the longer fatty acids attached that thus, overall, led to higher hydrophobicity. We did not find any difference between saturated (C18:0) and unsaturated fatty acids (C18:2). For that reason, we now only report for C18:0 modification. We expect that the current modification makes the particles suitable for application in polylactic acid that is hydrophobic, as tested in the next section. If successful, this most probably implies that modified ChNC are also suitable for application in other hydrophobic plastics such as polypropylene (PP) or polyethylene terephthalate (PET).

### 3.2. Nanocomposite Characterization

As a next step, nanocomposites were made, characterized, and compared to neat polylactic acid.

#### 3.2.1. Nanocrystal Dispersion in the PLA Matrix

We used scanning electron microscopy (SEM) to observe the dispersibility of ChNC in PLA (Figure 5). To be complete, PLA without ChNC is shown in Appendix A Figure A4. Nanocomposites with unmodified ChNC contained highly aggregated nanocrystals even >50 µm (Figure 5A,C). When the nanoparticles were washed before application, the number of small aggregates seemed less (Figure 5C). PLA samples containing modified ChNC mainly showed aggregates with a size around ~200 nm (Figure 5B,D,E). This indicates that surface modification facilitates ChNC dispersion. Bigger aggregates with a size > 20 µm were also observed, although they were much less abundant.

Figure 6 shows pictures of PLA containing 5 wt.% (modified) ChNC; Lab* color scores are provided to quantify color differences. Figure 7 shows film transmittance in the UV and visible light range.

As expected, PLA had a high transparency of 83.7% at wavelength 500 nm. The addition of 5 wt.% (modified) ChNC introduced a yellow-to-deep brown color, depending on the treatment used. This is in line with other studies that observed color formation upon extrusion [20]. Glucosamine-derived products, such as chitin, are known to undergo Maillard reactions at elevated temperatures, giving rise to a brownish color [26]. Interestingly, the presence of unmodified ChNC resulted in darker films compared to their modified counterparts. One possible explanation is the better dispersibility of modified ChNC in PLA (Figure 5B,D,E), which results in higher overall transparency. Alternatively, introduction of fatty acids to the ChNC’s surface may inhibit Maillard reactions.

Regardless of its treatment, the addition of ChNC reduced transmittance throughout the whole wavelength range measured (Figure 7), i.e., λ = 250–650 nm. For instance, at a wavelength of λ = 500 nm, the transmittance was 1.9% for PLA with unmodified unwashed ChNC (dark film) and 16.1% in the presence of ChNC-C8:0 (light film). This is an important clue for the development of food packaging materials that are less UV-transparent, making food products thus less prone to oxidation reactions and other reactions that are light-induced.

#### 3.2.2. Barrier Properties

Table 3 presents the water vapor transmission rate (WVTR) of neat PLA and nanocomposites containing 5 wt.% ChNC. Generally, the WVTR of the nanocomposites was ~7% lower than neat PLA, with the exception of nanocomposites containing washed ChNC that showed a high WVTR of ~260% compared to PLA. In the latter nanocomposites, substantial aggregation took place, most probably leading to ‘weak spots’ in the material, which was mitigated in the other nanocomposites. Improved barrier properties are commonly reported for nanocomposites [27,28,29], and many authors explain this by an increased tortuous diffusion path caused by nanoparticle addition, and this may also be the reason for the improved performance of our other nanocomposites in which the particles are much better dispersed.

#### 3.2.3. Mechanical Properties

Figure 8 shows the Young’s modulus (Figure 8A), maximum stress (Figure 8B), and elongation at break (Figure 8C) of neat PLA and the nanocomposites. The addition of ChNC particles results in a slightly higher Young’s modulus and slightly lower elongation at break. For instance, a Young’s modulus of 3030 ± 70 and 3154 ± 438 MPa was found for neat PLA and upon addition of 5 wt.% unwashed ChNC, respectively. It is good to point out that these differences are very small, and most probably insignificant. Others have reported increased maximum stress and Young’s modulus upon polysaccharide nanocrystal addition to PLA [20,30]. This is commonly explained by the formation of a percolation network of polysaccharide nanocrystals that gives rise to increased mechanical properties. The difference with our work is that, commonly, substantial amounts of plasticizers are used to facilitate ChNC dispersion, or PLA and ChNC are mixed with other plastics such as PBAT; this may not only facilitate nanoparticle dispersion, but also influence the mechanical film properties beyond what is possible within our experimental conditions.

## 4. Conclusions

Steglich esterification was successfully used to modify chitin nanocrystals, with fatty acids differing in carbon chain length and degree of saturation; covalent attachment was confirmed with FTIR and ^13^C NMR. We demonstrated that substitution of 2–4% influenced phase behavior greatly, with ChNC modification with the longest fatty acid leading to the highest hydrophobicity.

SEM observations suggested that modified ChNC dispersed better in the PLA matrix compared to their unmodified counterparts. This reduced brown color formation and improved transparency. Generally, the addition of ChNC provided high UV protection and improved barrier protection, which was without being at the expense of mechanical strength.

The prepared nanocomposites are relevant for application in, e.g., food, for which it can be expected that light-induced reactions will be slowed down considerably. The modified particles as such are also expected to be compatible with other hydrophobic polymers, and may contribute to development of other advanced packaging materials.

## Figures and Tables

**Figure 1 polymers-14-02619-f001:**
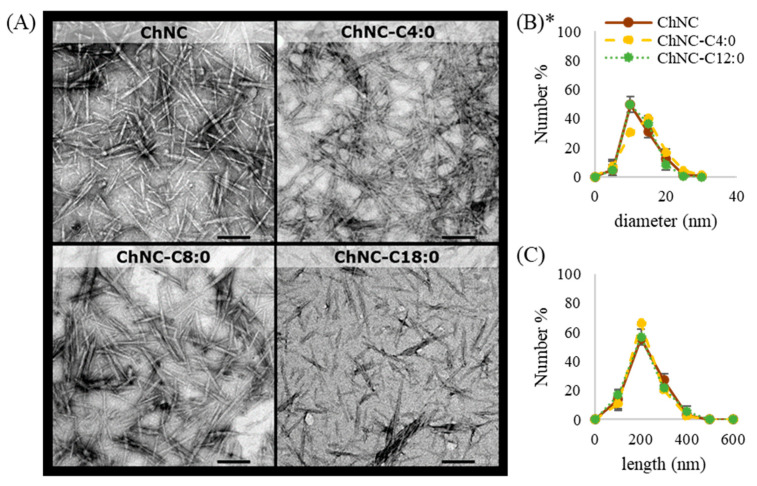
(**A**) Transmission electron microscopy images of ChNC as such and ChNC modified with C4:0, C8. (**B**) The particle diameter and (**C**) particle length distribution determined from TEM images. The error bars represent the standard deviation of three different regions; a total of 50 particles were measured per region. * It was not possible to determine the particle diameter and length of ChNC-C18:0 due to lower resolution of the pictures of this sample. By eye, this sample looked similar to the other three.

**Figure 2 polymers-14-02619-f002:**
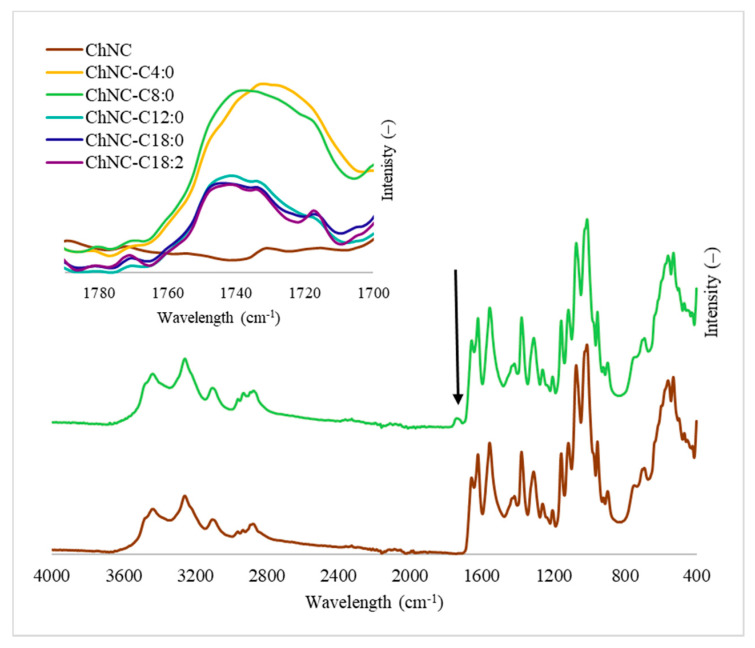
FTIR spectra of unmodified chitin nanocrystals and nanocrystals modified with fatty acids. The presence of the ester group at 1735–1750 cm^−1^ is highlighted in the inset; its intensity is dependent on the fatty acid of choice.

**Figure 3 polymers-14-02619-f003:**
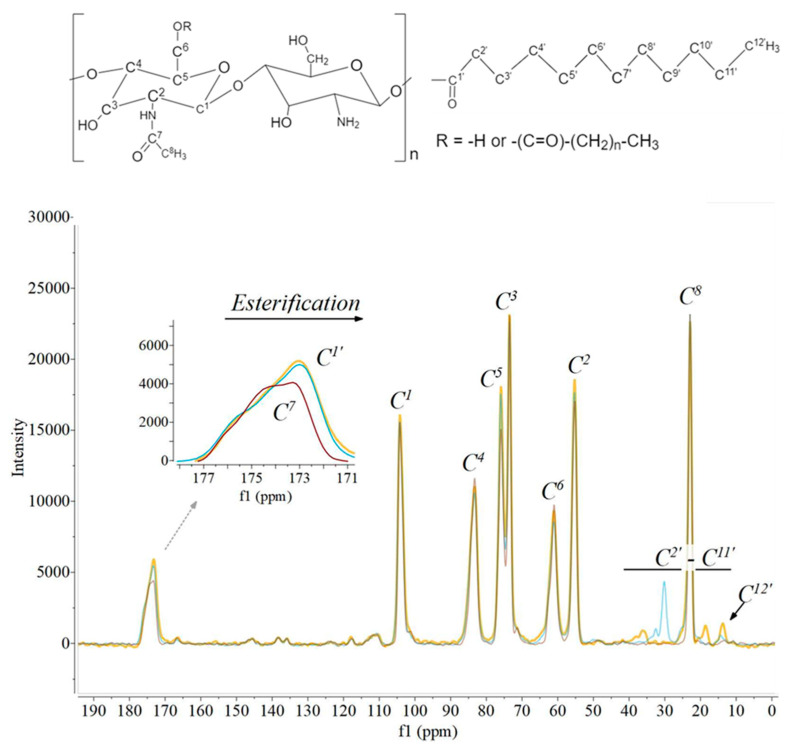
^13^C NMR spectra of ChNC (brown), ChNC-C4:0 (yellow), and ChNC-C12:0 (blue). The chemical structure of chitin and the fatty esters is provided, where n can be 2, 6, 10, or 16 depending on the fatty ester considered.

**Figure 4 polymers-14-02619-f004:**
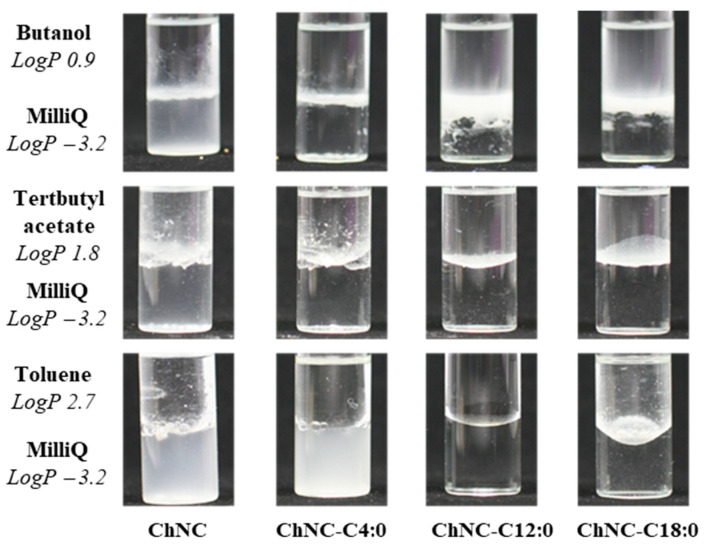
Phase behavior of (un)modified ChNC in two-phase systems, indicative of their hydrophobicity.

**Figure 5 polymers-14-02619-f005:**
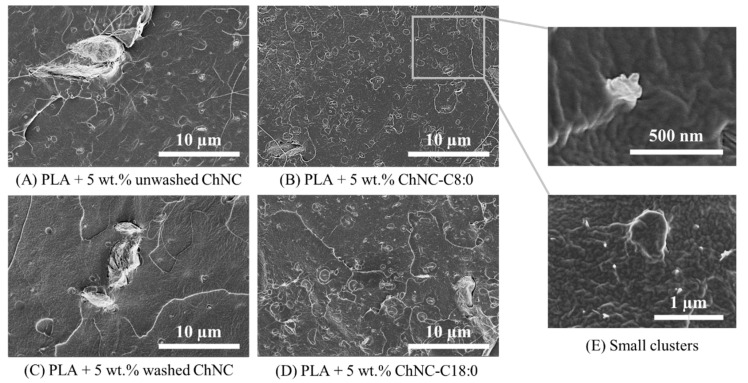
Scanning electron microscopy pictures of PLA containing (**A**) 5 wt.% unwashed ChNC, (**B**) 5 wt.% ChNC-C8:0, (**C**) 5 wt.% washed ChNC, (**D**) 5 wt.% ChNC-C18:0. (**E**) Samples containing ChNC-C8:0 and ChNC-C18:0 had many small aggregates with a size of ~500 nm.

**Figure 6 polymers-14-02619-f006:**
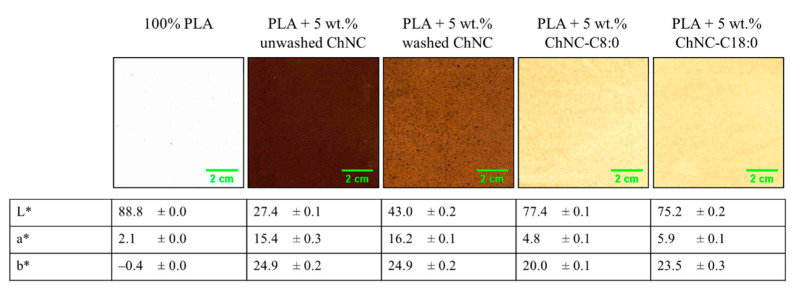
Pictures of nanocomposites containing 5 wt.% (modified) chitin nanocrystals and their corresponding L*, a*, and b* scores.

**Figure 7 polymers-14-02619-f007:**
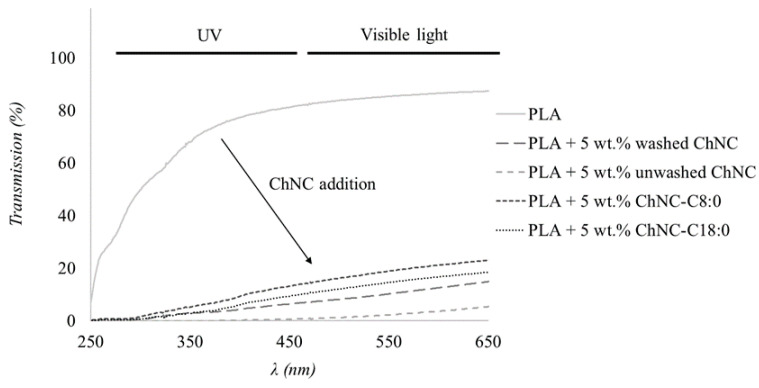
Transmittance of PLA films containing 5 wt.% (modified) ChNC. Wavelengths in UV and visible ranges are highlighted.

**Figure 8 polymers-14-02619-f008:**
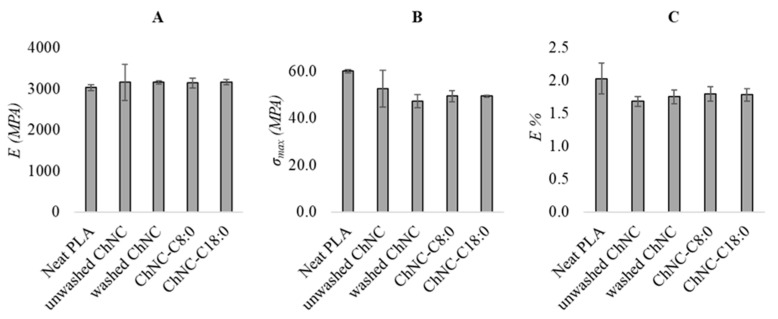
Young’s modulus (**A**), maximum stress (**B**), and elongation at break (**C**) of neat PLA and nanocomposites containing unmodified (unwashed and washed) and modified chitin nanocrystals (ChNC-C8:0 and ChNC-C18:0).

**Table 1 polymers-14-02619-t001:** The ^13^C NMR chemical shift of C^7^ for unmodified ChNC and C^1′^ for modified ChNC, including the peak area compared to C^1^.

	Chemical Shift C^7^ or C^1′^	Area Compared to C^1^
ChNC	174.9	0.35
ChNC-C4:0	173.0	0.65
ChNC-C8:0	172.9	0.65
ChNC-C12:0	173.1	0.62
ChNC-C18:0	172.9	0.65
ChNC-C18:2	173.0	0.63

**Table 2 polymers-14-02619-t002:** The degree of substitution (DS%), degree of acetylation (DA%), and ζ-potential at pH = 5.0 of (modified) chitin nanocrystals.

Sample	DS%	DA%	ζ-Potential (mV)
ChNC	(⎼)	74.5	35.3 ± 0.5
ChNC-C4:0	2.6 ± 0.2	78.4	33.9 ± 0.6
ChNC-C8:0	3.9 ± 0.0	78.5	34.7 ± 0.3
ChNC-C12:0	2.2 ± 0.1	71.3	35.0 ± 1.0
ChNC-C18:0	1.7 ± 0.0	60.7	35.7 ± 0.5
ChNC-C18:2	2.0 ± 0.5	62.9	34.3 ± 0.2

**Table 3 polymers-14-02619-t003:** Water vapor transmission rate (WVTR) of PLA nanocomposites containing 5 wt.% (modified) ChNC.

Sample	WVTR (g/m^2^.day) *	WVTR Compared to PLA
PLA	38.2 ± 2.2	-
PLA + 5 wt.% washed ChNC	99.7 ± 73.3	+261.4%
PLA + 5 wt.% unwashed ChNC	35.6 ± 0.9	−6.7%
PLA + 5 wt.% ChNC-C8:0	35.6 ± 0.5	−6.3%
PLA + 5 wt.% ChNC-C18:0	34.6 ± 2.3	−9.4%

* 100 µm thickness, 23 °C, 85% gradient RH.

## Data Availability

Data available on request due to technical limitations.

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
