# Peer review of "Chitin Nanocrystal Hydrophobicity Adjustment by Fatty Acid Esterification for Improved Polylactic Acid Nanocomposites"

_polymers, 2022, doi:10.3390/polym14132619_

Round 1

Reviewer 1 Report

The manuscript deals with the fatty acid esterification of Chitin nanoparticles and their use for the production of PLA-based nanocomposites.

Title: Why the title refers to “chitin nanoparticles”, while the other parts of the manuscript refers to “chitin nanocrystals”?

Introduction: The rationale about formulating a PLA nanocmposites with these derivates should be highlighted at the end of the introduction

Methods: Paragraph 2.3 How was controlled the thickness of the prepared nanocomposite?

Paragraph 2.5.3. How light trasmittance across the nanocomposite was measured? How nanocomposites were placed in the sampe holder of the UV instrument?

Paragraph 2.5.5 Which is the geometry and dimensions of the samples tested by tensile strenght measurements?

In the results section, detailed description of 13C-NMR for each esterification should be provided. A further chemical characterizion of the conjugates could be also useful.

Line 269 and Table 1 How the degree of acetylation was determined?

Reviewer 2 Report

The paper is devoted for chitosan nanoparticles and composites with these inclusions prperation and characterization. The topic is generally interesting, however the paper contains unexplained places (below) and need major revisions.

In parts 2.2 and 2.3 it described samples preparation procedure. Why samples were prepared under exactly such conditions? I not find any reference or explanation. The preparation procedure has the impact on particles and composites properties presented in Figs. 1-8?

Figure 7, please explain why transmission decreases with ChNC? What optical properties have ChNC?

Figure 8 should be more commented.

Conclusions should be more informative.

English need minor revisions. Misprints should be corrected, for example lines 47-58 cm-1.

Round 2

Reviewer 1 Report

The manuscript is suitable for publication

Reviewer 2 Report

Authors make proper corrections according to reviewer remarks and I suggest to publish the paper as it is.